# Influence of Metal Implants on Quantitative Evaluation of Bone Single-Photon Emission Computed Tomography/Computed Tomography

**DOI:** 10.3390/jcm11226732

**Published:** 2022-11-14

**Authors:** Keisuke Oe, Feibi Zeng, Takahiro Niikura, Tomoaki Fukui, Kenichi Sawauchi, Tomoyuki Matsumoto, Munenobu Nogami, Takamichi Murakami, Ryosuke Kuroda

**Affiliations:** 1Department of Orthopaedic Surgery, Kobe University Graduate School of Medicine, Chuo-ku, Kobe 650-0017, Japan; 2Department of Radiology, Kobe University Graduate School of Medicine, Chuo-ku, Kobe 650-0017, Japan; 3Biomedical Imaging Research Center, University of Fukui, Fukui 910-1193, Japan

**Keywords:** bone, computed tomography-based attenuation correction, single-photon emission computed tomography/computed tomography, metal implant, standardized uptake value

## Abstract

When visualizing biological activity at nonunion sites by the radioisotopes, gamma rays are more attenuated if metal implants are placed in the bone. However, the effects of various implant types and their placement on gamma ray attenuation in quantitative evaluation remain unknown. To elucidate these effects, we created a phantom that simulated the nonunion of the femur in this study. The count of gamma rays was measured by single-photon emission computed tomography/computed tomography (SPECT/CT) while considering CT-based attenuation correction (CTAC), metal implant placement, type (intramedullary nail or plate), and position. The count differed significantly with and without CTAC and with and without implants (both types) under CTAC. Significantly different counts were observed between the intramedullary nail and plate placed contralaterally to the lesion (i.e., non-lesion side). No significant difference was observed between the intramedullary nail and plate on the lesion side or between plates on the non-lesion and lesion sides. The measured standardized uptake value (SUV) was closer to the true SUV with CTAC than without. Moreover, the count was higher with implants than without. However, even with implants, it was lower than the actual count, indicating the absence of overcorrection. Implant type and position do not seem to influence the count.

## 1. Introduction

Nonunion occurs in approximately 5% of all fractures [1,2]. The various causes of nonunion can be largely categorized into biological factors and mechanical factors [3,4,5]. Generally, the cause of nonunion is determined by taking X-rays of the fractured site. The presence of marked callus formation around the nonunion site typically indicates biological activity; however, in cases without callus formation, it is very difficult to determine the existence of biological activity using only X-ray findings. We previously quantified nonunion fracture repair in terms of the standardized uptake value (SUV) of single-photon emission computed tomography (SPECT) in hypertrophic and non-hypertrophic nonunions [6]. We found high SUVs in cases of non-hypertrophic nonunion, which usually exhibit poor biological activity. This finding suggests the existence of biological activity. Therefore, we believe that bone SPECT quantification will be useful in autologous bone grafting during nonunion surgery [6,7].

In recent years, SPECT coupled with CT (SPECT/CT) has been used to obtain highly sensitive and specific images when diagnosing bone lesions. Moreover, as fewer metal artifacts are formed during SPECT/CT imaging than during magnetic resonance imaging, orthopedic surgeons find SPECT/CT convenient considering that they often use metal implants as fixation tools for treating fractures and creating artificial joints [8,9,10,11,12]. Due to the aging society in Japan and several other countries, the number of patients with peri-implant fractures along with the number of nonunion fractures that occur after peri-implant fractures is increasing. Most cases of nonunion fractures are surgically treated by inserting metal implants in and around the fracture site. Therefore, when a radioisotope is administered during SPECT/CT, the emitted gamma rays are attenuated by these metal implants with increased duration of exposure. For this reason, the attenuation of gamma rays caused by metal implants during quantitative evaluation is usually corrected using CT-based attenuation correction (CTAC) [13,14]. CT values are subjected to CTAC using a function that is inherent to SPECT/CT equipment. Because the treatment method (e.g., autologous bone grafting) is chosen based on the biological activity at the nonunion site and the known therapeutic effects of anticancer drugs for bone tumors, more accurate quantitative values are required. Therefore, CTAC is an indispensable tool as it can help improve the accuracy of such values.

Metal implants are of various types, such as plates, intramedullary nails, and artificial joints and composed of different materials. However, no studies have reported the effects of CTAC and the type and placement position of the implants on bone. Therefore, we investigated these effects using a phantom that imitated the nonunion of a femur (Figure 1).

## 2. Materials and Methods

### 2.1. Phantom and Implants

In this study, the material used to construct the phantom (NMP Business Support Co., Ltd., Sanda, Japan) was synthetic resin. The structure was 400 mm in length and 40 mm in diameter. The phantom was shaped like a femur and had an R1400 curve. Two parts that mimicked lesions (nonunion) of 3000 mm^3^ and 2000 mm^3^ were created on the proximal and distal portions of the phantom, with each portion comprising 1/3rd of the phantom (Figure 2a) [15,16].

Pertechnetate (^99m^TcO_4_^−^) (MEDITEC^®^ Generator; Nihon Medi-Physics, Tokyo, Japan) was the radioisotope used. The concentration of ^99m^TcO_4_^−^ injected into the lesion was 300 kBq/mL, while the concentration injected into the bone, which serves as the background, was 50 kBq/mL as done in a previous study [13]. The half-life of ^99m^TcO_4_^−^ is 6.01 h.

The implants used were a femoral plate with a length of 350 mm (NCB Periprosthetic Distal Femur Plate; Zimmer Biomet, Warsaw, IN, USA) and a femoral nail with a diameter of 9.3 mm and length of 420 mm (Natural nail GT-femoral; Zimmer Biomet, Warsaw, IN, USA) (Figure 2b–d). Both implants were constructed using titanium alloy (Ti6Al4V: Ti-6Al-4V Alloy, Wrought ELI Grade, 64WF Metal Alloy/Ti-6Al-4V Alloy, ISO 5832-3/ASTM F-136; aluminum 5.50 to 6.50, vanadium 3.50 to 4.50, carbon 0.08 max, nitrogen 0.05 max, oxygen 0.13 max, hydrogen 0.012 max, iron 0.25 max, titanium balance).

### 2.2. SPECT/CT

SPECT/CT imaging was performed on an Optima NM/CT 640 scanner system (GE Healthcare, Tokyo, Japan), which consists of a dual-head γ-camera and CT scanner. The SPECT scan was acquired using a low-energy, high-resolution collimator at a photoenergy peak of 140 keV for ^99m^TcO_4_^−^ with a matrix of 128 × 128, pixel size of 4.4 mm, magnification of 1, and total of 60 projections (30 steps) over 360° with a dwelling time of 15 s/step. All SPECT images were reconstructed using three-dimensional ordered subset expectation maximization with six iterations, 10 subsets, and a Butterworth filter (order: 10, cutoff: 0.50 cycle/cm). Scatter correction was performed by applying dual energy windows with different sub-energy window widths (main: 140.5 keV ± 10%, sub: 120 keV ± 5%). Transaxial slices were reconstructed with no attenuation correction (NC) or CTAC, which are functions provided by the scanner system. CT images were acquired on an Optima NM/CT 640 scanner system and co-registered with emission data for CTAC and NC. CT scanning was performed at 120 kV and 10 mA with a tube rotation time of 1.0 s, pitch of 1.75, transverse field of view having a diameter of 50 cm, matrix of 512 × 512, and slice thickness of 2.5 mm.

### 2.3. SUV Measurements

The quantitative SPECT parameters were calculated using the GI-BONE software (AZE, Tokyo, Japan). The ^99m^TcO_4_^−^ uptake was quantitatively analyzed by calculating the SUV using the following equation:

SUV = (tissue radioactive concentration/voxel volume)/(injected radioactivity/body weight), where tissue radioactive concentration is the value obtained by multiplying the SPECT counts and Becquerel calibration factor, which was determined by scanning the cylindroid phantom filled with a known concentration of radioisotope. Various SUV parameters were calculated using GI-BONE. The maximum value for SUV (SUV_max_) = (maximum radioactivity/voxel volume)/(injected radioactivity/body weight). The mean value for SUV (SUV_mean_) = (total radioactivity/volume of interest (VOI))/(injected radioactivity/body weight). The peak value of SUV (SUV_peak_) represents the average SUV obtained within a 1 cm^3^-sphere representing the region of interest centered around the highest voxel of the target area.

The VOI was defined as a cylinder with a diameter of 9 mm and height of 17 mm, considering the limit of the spatial resolution of SPECT [16]. An orthopedic trauma surgeon and a radiologist identified and measured the uptake site by observing the CT and SPECT images.

### 2.4. Experimental Procedure

First, 50 kBq/mL and 300 kBq/mL of ^99m^TcO_4_^−^ were injected into the phantom at the sites where the bone and two lesions were simulated, respectively. Next, the phantom was scanned using SPECT/CT in the following sequence: (1) with the plate on the lesion side, (2) with the plate on the non-lesion side, (3) with the intramedullary nail installed, and (4) with the implant removed (Figure 3 and Figure 4).

### 2.5. Evaluation

First, the number of gamma rays captured by the gamma camera were counted with and without performing CTAC. Next, the differences between the counts in the presence and absence of the implant under CTAC, with different types of implants (plate or intramedullary nail), and with different implant installation positions (intramedullary position, lesion side, or non-lesion side) were evaluated by plotting the count profile curve.

The calculated ideal SUVs of the lesion on the phantom were compared with the measured SUVs. The true SUV was calculated on the basis of the subject’s radioactive concentration, dosed radioactivity, and weight.

### 2.6. Statistical Analysis

In this study, statistical analysis was performed using the Wilcoxon signed rank test. *p* < 0.05 was considered to be statistically significant.

## 3. Results

There was a significant difference between the counts in the presence and absence of CTAC. The count profile curves had higher peaks in the presence of CTAC in all cases, regardless of the presence or location of the implant (Figure 5).

There was a significant difference in the counts with the implants (lesion side, non-lesion side, and inside) and without them when CTAC was performed (*p* < 0.001 in all scenarios compared). The counts when the plate was installed on the non-lesion side was significantly different from when the intramedullary nail was installed inside the phantom. In the presence of CTAC, no significant differences between the counts when the plate was installed at the non-lesion side and lesion side were observed as well as when the plate was installed on the lesion side and the intramedullary nail was installed inside (Figure 6 and Table 1).

The ideal SUV was calculated as 7.43. SPECT/CT examination with CTAC showed that the SUV_max_, SUV_peak_, and SUV_mean_ were 5.50 ± 0.18, 4.63 ± 0.17, and 4.95 ± 0.42, respectively, whereas the same parameters measured without CTAC were 3.79 ± 0.21, 3.20 ± 0.18, and 3.42 ± 0.40, respectively. These are the average SUVs collected while investigating each of the eight scenarios with large and small lesions (Figure 7). No significant differences between the SUVs based on the size of the lesion were observed.

## 4. Discussion

To date, few reports have investigated the effects of CTAC and implants on lesions around total hip arthroplasty [14,17]. The authors of these reports have stated that further correction is needed because the SUV will be overcorrected if the bone lesion is nearby. However, the difference in counts by implant type, implant placement position, and CTAC of SUV upon exposure to gamma rays has not been determined before. This study is the first to investigate the effects of the type and location of metal implants on the quantitative analysis of bone lesions using SPECT/CT.

To summarize the results in relation to CTAC, the count was significantly lower if CTAC was not performed, indicating the necessity of performing CTAC. Next, regarding the presence of the implant, the count was higher with the implant than that without the implant. Therefore, it is possible that excessive CTAC had occurred. However, because the true SUV was not exceeded even with CTAC in this study, it was concluded that CTAC should be performed along with implant installation.

The counts when the plate was placed inside the phantom (intramedullary nail) were significantly different from when it was placed at the non-lesion side. However, no significant difference in the count was observed between the phantom when the plate was installed at the lesion side and when it was installed at the non-lesion side. Therefore, in this study, the count was not considered to be related to the type or location of the implant.

The future prospects of this study are myriad. In this study, we measured the counts with one implant at a time. However, in clinical practice, multiple implants are inserted [18,19,20,21,22]. Therefore, the effects of inserting multiple implants at the same time will be investigated in the future. Likewise, the effects of the material of the implant can be studied further. Herein, only implants made of titanium alloy were studied, but we intend to perform additional studies on other types of metals such as stainless steel and cobalt chrome alloy. Furthermore, quantitative evaluation using more accurate SUVs is expected to provide useful information for selecting suitable treatments and drugs, which will pave the way for personalized medicine in the future.

Our study has a few limitations. First, because bone SPECT/CT has low spatial resolution, it cannot accurately evaluate small objects of size 17 mm or less. Second, a phantom that simulated the femur was used in this study as a model specialized for bone lesions and implants; thus, our study did not consider soft tissues such as muscles, vessels, nerve, and skin. As the objective of the study was to evaluate the relationship between bone lesions and implants, it was considered appropriate to use a phantom with a simple structure.

## 5. Conclusions

In this study, we investigated several scenarios that may affect the results of bone SPECT/CT. The measured SUVs were found to be closer to the true SUV when CTAC was performed than when it was not performed. Moreover, the number of gamma rays (i.e., counts) was higher when the implant was inserted than when the implant was not inserted. However, even with implants, the count was not higher than the actual count, indicating that overcorrection did not occur. In addition, the type of implant and the installation position do not seem to influence the quantitative analysis of the bone lesion using SPECT/CT. This study on the necessity of CTAC and the effects of metal implants is of great significance for nuclear medicine researchers, clinicians, and patients not only in orthopedics, but also in the field of oncology, where quantitative evaluation of SPECT/CT is emphasized.

## Figures and Tables

**Figure 1 jcm-11-06732-f001:**
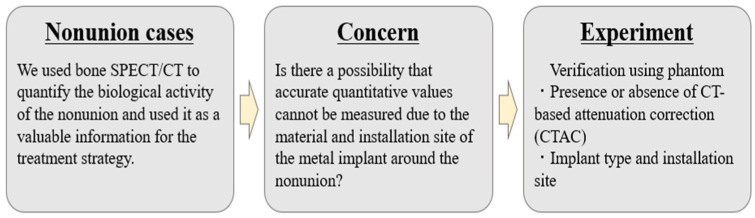
Comprehensive schema of research.

**Figure 2 jcm-11-06732-f002:**
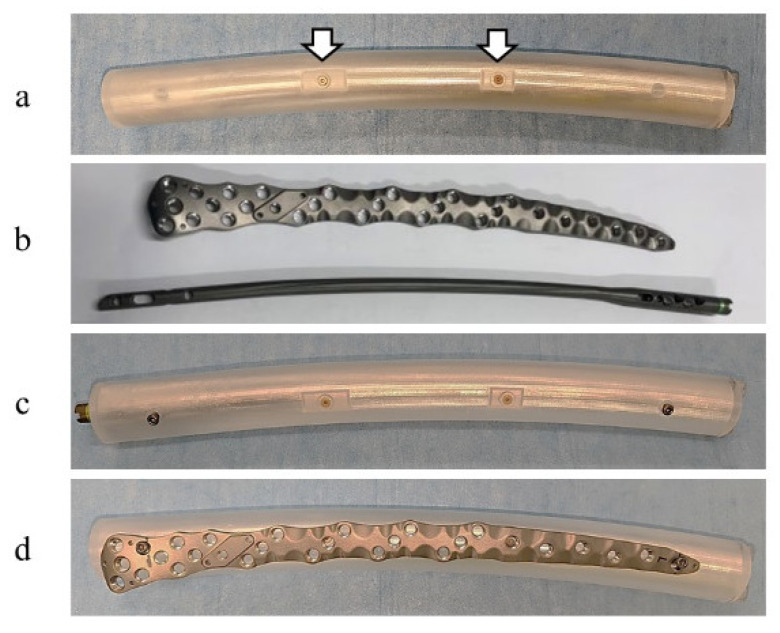
Photographs of the phantom and implants used in this study. (**a**) Two lesions (white arrows) were present in the phantom, and the structure was constructed such that an intramedullary nail could be installed in the center. (**b**) The implants at the top and bottom of this panel are the plate and intramedullary nail, respectively. (**c**) The installed intramedullary nail and (**d**) plate are shown.

**Figure 3 jcm-11-06732-f003:**
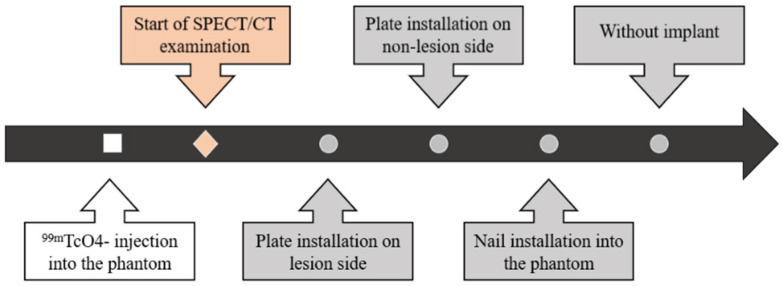
Flowchart of the imaging procedure. First, ^99m^TcO_4_^−^ was injected into the phantom. Next, the phantom was placed on the SPECT/CT imaging table and visualized after each modification, which were performed in the following order: plate installation on the phantom at the lesion side, plate installation contralateral to the lesion (i.e., on the non-lesion side), intramedullary nail installation, and implant removal.

**Figure 4 jcm-11-06732-f004:**
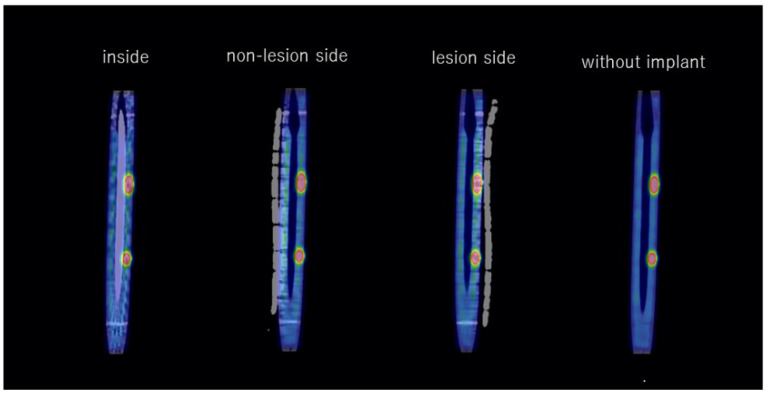
Fused coronal images (SPECT and CT) with and without the implant.

**Figure 5 jcm-11-06732-f005:**
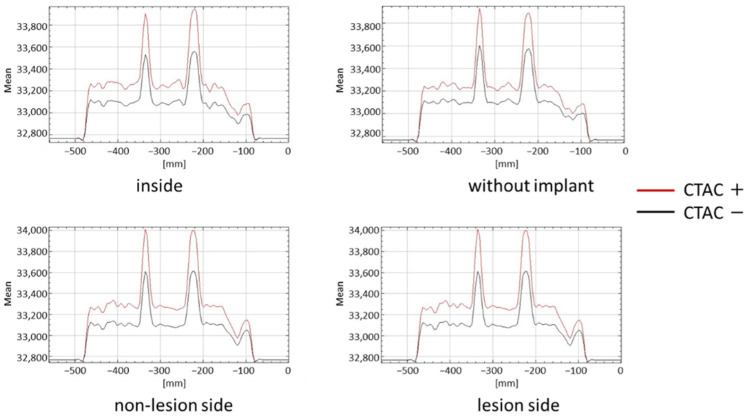
Quantification of lesions according to implant type and installation position with and without CTAC.

**Figure 6 jcm-11-06732-f006:**
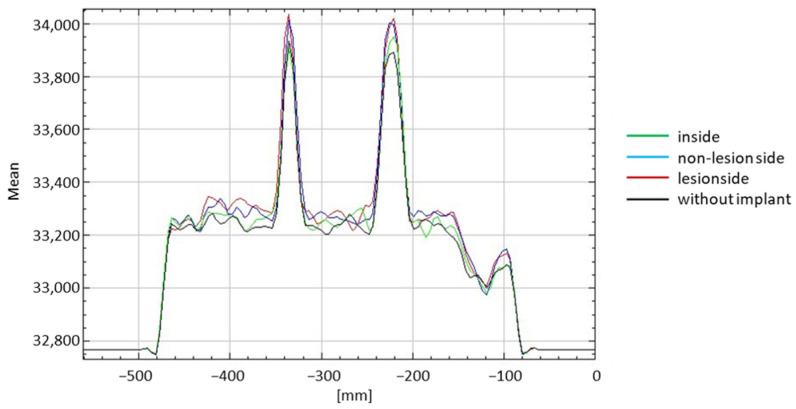
Summarized evaluation of each scenario investigated with CTAC.

**Figure 7 jcm-11-06732-f007:**
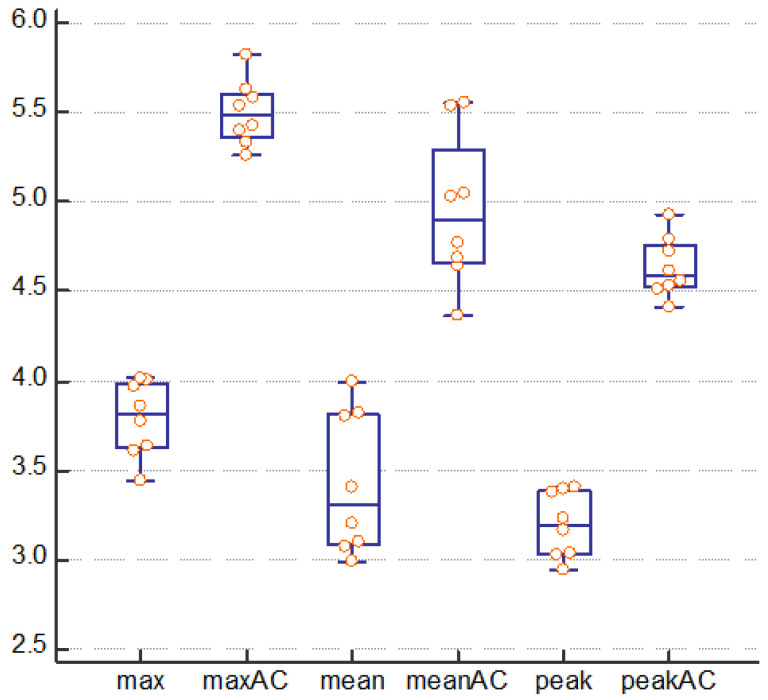
SUV after SPECT/CT examination with and without performing CTAC.

**Table 1 jcm-11-06732-t001:** Tabulated results of the evaluation of each scenario investigated with CTAC.

	Inside	Lesion Side	Non-Lesion Side
Without implant	*p* < 0.001	*p* < 0.001	*p* < 0.001
Lesion side	*p* = 0.087		
Non-lesion side	*p* < 0.001	*p* = 0.087	

## Data Availability

The datasets generated and/or analyzed during the current study are available from the corresponding author on reasonable request.

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
