# Peer review of "Influence of Metal Implants on Quantitative Evaluation of Bone Single-Photon Emission Computed Tomography/Computed Tomography"

_jcm, 2022, doi:10.3390/jcm11226732_

Round 1

Reviewer 1 Report (Previous Reviewer 1)

All points of criticism have been adressed as long as I am concerned. Very good work on that part.

Author Response

Thank you very much for your kind review.

Reviewer 2 Report (Previous Reviewer 2)

1. Authors should first make a clear schematic illustration for viewers to comprehend the scope of the manuscript.

2.Pertechnetate ( 99mTcO4 - ) was the radioisotope used. ( Manufacturer details?)

3.Pertechnetate ( 99mTcO4 - ) was injected only ones in the starting, does the does remain the same with time or it is cleared by the system? Repeated doses at intervals could have clearly established this clearly.

4. No parameters details of GI-BONE software?

5. It was concluded that CTAC 212 should be performed along with implant installation. what is the experimental basis of this statement?

6. Funding information should be in detail with the funding number for proper acknowledgment.

Author Response

1. Authors should first make a clear schematic illustration for viewers to comprehend the scope of the manuscript.

Response: Thank you for your comment. We added the comprehensive schematics. However, we couldn’t draw an illustration, so we made a document. Please refer to Figure 1.

2. Pertechnetate (99mTcO4 - ) was the radioisotope used. (Manufacturer details?)

Response: Thank you for your comment. We added the Manufacturer details. Please refer to line 81.

3. Pertechnetate (99mTcO4 -) was injected only ones in the starting, does the does remain the same with time or it is cleared by the system? Repeated doses at intervals could have clearly established this clearly.

Response: Thank you for your comment. Pertechnetate is not removed sequentially. The same radioisotope is generally used in such experiments.

4. No parameters details of GI-BONE software?

Response: Thank you for your comment. The parameters provided by the software are as described in the Materials and Methods (2.3. SUV measurements).

5. It was concluded that CTAC 212 should be performed along with implant installation. what is the experimental basis of this statement?

Response: Thank you for your comment. The experiment yielded the result the count was significantly lower if CTAC was not performed and indicated the necessity of performing CTAC.

6. Funding information should be in detail with the funding number for proper acknowledgment.

Response: I contacted the foundation, but they told me there was no funding number. Instead, I’ll send you the URL of the foundation I got the grant.

https://social.ja-kyosai.or.jp/jibaiseki/ 

Reviewer 3 Report (Previous Reviewer 3)

thank you for the changes. My concerns have been addressed.

Author Response

Thank you very much for your kind review. 

Reviewer 4 Report (Previous Reviewer 4)

The authors modified the paper according to the recommendations.

Author Response

Thank you very much for your kind review. 

This manuscript is a resubmission of an earlier submission. The following is a list of the peer review reports and author responses from that submission.

Round 1

Reviewer 1 Report

It would be of advantage for the understanding of the reader if the authors would describe CTAC and its current use as examples, in the introduction section.

Author Response

Reviewer #1:

It would be of advantage for the understanding of the reader if the authors would describe CTAC and its current use as examples, in the introduction section.

Response: Thank you for your comment. We have added an example of the use of CTAC in the Introduction section. Please refer to lines 55-67.

Reviewer 2 Report

1. A systematic review of various implant types and their effect on degradation can immediately catch the viewer's interest.
2. From the abstract the article could not be comprehended.

3. Schematic representation is lacking.

4. Overall the manuscript lacks the novelness and innovativeness for publication

Author Response

Reviewer #2:

  1. A systematic review of various implant types and their effect on degradation canimmediately catch the viewer's interest.

Response: Thank you for your comment. We would like to continue our research and take on the challenge of a systematic review in future.

  1. From the abstract the article could not be comprehended.

Response: Thank you for your comment. We have revised the content of the abstract for further clarity. Please confirm that it is now acceptable.

  1. Schematic representation is lacking.

Response: Thank you for your comment. We have changed the schematics. Please refer to Figure 2.

  1. Overall the manuscript lacks the novelness and innovativeness for publication

Response: To the best of our knowledge, this is the first study that considers CT attenuation correction in osteosynthesis implants. As a result of conducting this study, we found that gamma ray counts in lesions required CTAC regardless of the presence or absence of implants and that it had no relationship with the type or placement position of implants. We consider these findings to be novel.

Reviewer 3 Report

This is an interesting and useful study, but I think additional clarity is needed. 

It seemed that the abstract did not agree with or reflect the findings as enumerated in the results or discussion. Is your abstract accurate? Perhaps it is related to language use?  I would also be interested in the scale, accuracy,  calibration and standardization of the CTAC protocol,  which was simply mentioned as a function of the machine used. Is the protocol the same for all machines? Do you have a reference for how this is handled? I am an orthopaedic surgeon so my familiarity with this function is limited. 

Lline 81--I do not understand the meaning. 

line 83 and 84-- maybe language could be clearer?

Your first statement suggests that implants cause attenuation so CTAC is recommended. ( The English is not completely clear here-- Maybe I did not understand?) Do you have and explanation for why the lesion count was higher with implant present, if it is expected to cause attentuation? This does not make much sense to me. If it is thought to be related to scatter, maybe that should be postulated.

Is there a way to do a baseline comparison or control of phantom without lesions with and without CTAC?--I did not see where this was done. Perhaps this could also be done with no lesions but with and without implants. I feel more control is needed. 

Since you measured phantom with lesions but no implants, with and without CTAC, If CTAC is needed to compensate for implants, perhaps a ratio could be generated or at least help to determine the level of correction needed? Is CTAC protocol  adjustable?

Author Response

Reviewer #3:

This is an interesting and useful study, but I think additional clarity is needed. It seemed that the abstract did not agree with or reflect the findings as enumerated in the results or discussion. Is your abstract accurate? Perhaps it is related to language use?  I would also be interested in the scale, accuracy, calibration and standardization of the CTAC protocol, which was simply mentioned as a function of the machine used. Is the protocol the same for all machines? Do you have a reference for how this is handled? I am an orthopaedic surgeon so my familiarity with this function is limited. 

Response: Thank you for your comments. We have corrected the missing parts of the Abstract. We think that it is now clearer. We suspect a possibility that measurement errors may occur between SPECT/CT machines. However, in this survey, we used the same model of the machine, and the protocol was set on the same time axis as this survey; hence, we believe that there was no measurement error.

line 81 I do not understand the meaning. 

Response: This shows the composition of the titanium alloy. If you do not recommend incorporating these sentences, we can delete these.

line 83 and 84 maybe language could be clearer?

Response: We have revised the sentences for further clarity.

Your first statement suggests that implants cause attenuation, so CTAC is recommended. (The English is not completely clear here-- Maybe I did not understand?) Do you have and explanation for why the lesion count was higher with implant present, if it is expected to cause attenuation? This does not make much sense to me. If it is thought to be related to scatter, maybe that should be postulated.

Response: Thank you for your comments. Certainly, the scattered rays caused by the implants may contribute to the result. It is difficult to evaluate the effect of scattering, so we used the same scattering correction for all the scans in this study to equalize conditions as much as possible.

Is there a way to do a baseline comparison or control of phantom without lesions with and without CTAC? I did not see where this was done. Perhaps this could also be done with no lesions but with and without implants. I feel more control is needed. 

Response: Thank you for your comments. Regarding the baseline of the lesion-free phantom, considering the resolution of SPECT/CT, a position more than 20 mm away from the lesion does not affect the analysis; hence, it can be regarded as a lesion-free phantom. Therefore, we did not perform measurements with a lesion-free phantom.

Since you measured phantom with lesions but no implants, with and without CTAC, If CTAC is needed to compensate for implants, perhaps a ratio could be generated or at least help to determine the level of correction needed? Is CTAC protocol adjustable?

Response: Thank you for your comments. We think that it is theoretically possible if we increase the number of cases to be examined; however, the purpose of this research was not to calculate the ratio. Thus, this will be a subject for future research.

Reviewer 4 Report

The paper presented for the review is aimed to assess the effects of various implant types and their placement on gamma radiation effects from the different types and sizes of imitated lesions on a phantom that simulated the femur.  The recent achievements in general management of primary and metastatic tumours as well as the successful orthopaedic treatment of the concomitant bony lesions lead to the new challenges such as the necessity of the comprehensive analysis of the data from the specific investigations which are sensitive to the pre-existing conditions (like metal implants). Radioisotope administration results in attenuation of gamma rays depending on the presence of metal implants in bones. Therefore, attenuation of gamma rays during quantitative evaluation is often corrected to account for implants.  The authors conducted an elegant study of the interrelations of the obtained data from the phantoms simulating the long tubular bones with the imitation of the bone lesions with the direct injection  of the radioisotope. The count and standardized uptake value were evaluated using single-photon emission computed tomography/computed tomography while considering CT-based attenuation correction, metal implant placement, type (intramedullary nail and plate), and position. As it was hypothesised the presence of the neighbouring massive implant did change the results of the scanning. In the series of the simulated experiments the authors demonstrated the difference of the appearance of this phenomenon depending on the type of the implant. Significantly different counts were observed between intramedullary nail and plate placed at the non-lesion side. No difference was observed at different lesion sizes. It is an interesting and important information which should be taken into account when assessing the results of the scanning in presence of the metal implants. When scanning with implants, it was lower than the actual count, indicating the absence of overcorrection. The results of the study can be assumed for the future development of the specific recalculation tool for the more precise assessment of the results of the scanning. 

Author Response

Reviewer #4:

The paper presented for the review is aimed to assess the effects of various implant types and their placement on gamma radiation effects from the different types and sizes of imitated lesions on a phantom that simulated the femur.  The recent achievements in general management of primary and metastatic tumors as well as the successful orthopaedic treatment of the concomitant bony lesions lead to the new challenges such as the necessity of the comprehensive analysis of the data from the specific investigations which are sensitive to the pre-existing conditions (like metal implants). Radioisotope administration results in attenuation of gamma rays depending on the presence of metal implants in bones. Therefore, attenuation of gamma rays during quantitative evaluation is often corrected to account for implants.  The authors conducted an elegant study of the interrelations of the obtained data from the phantoms simulating the long tubular bones with the imitation of the bone lesions with the direct injection of the radioisotope. The count and standardized uptake value were evaluated using single-photon emission computed tomography/computed tomography while considering CT-based attenuation correction, metal implant placement, type (intramedullary nail and plate), and position. As it was hypothesized the presence of the neighboring massive implant did change the results of the scanning. In the series of the simulated experiments the authors demonstrated the difference of the appearance of this phenomenon depending on the type of the implant. Significantly different counts were observed between intramedullary nail and plate placed at the non-lesion side. No difference was observed at different lesion sizes. It is an interesting and important information which should be taken into account when assessing the results of the scanning in presence of the metal implants. When scanning with implants, it was lower than the actual count, indicating the absence of overcorrection. The results of the study can be assumed for the future development of the specific recalculation tool for the more precise assessment of the results of the scanning. 

Response: Thank you very much for your kind review. We would like to continue our research so that our findings could be useful in clinical practice.
